# Characterization of Promising Cytotoxic Metabolites from *Tabebuia guayacan* Hemsl.: Computational Prediction and In Vitro Testing

**DOI:** 10.3390/plants11070888

**Published:** 2022-03-26

**Authors:** Seham S. El-Hawary, Rabab Mohammed, Marwa A. Taher, Sameh Fekry AbouZid, Mostafa A. Mansour, Suliman A. Almahmoud, Bader Huwaimel, Elham Amin

**Affiliations:** 1Department of Pharmacognosy, Faculty of Pharmacy, Cairo University, Giza 12613, Egypt; seham.elhawary@yahoo.com; 2Department of Pharmacognosy, Faculty of Pharmacy, Beni-Suef University, Beni-Suef 62514, Egypt; rmwork06@yahoo.com (R.M.); sameh.zaid@pharm.bsu.edu.eg (S.F.A.); 3Department of Pharmacognosy, Faculty of Pharmacy, Nahda University, Beni-Suef 62521, Egypt; marwa.taher@nub.edu.eg; 4Department of Pharmacognosy, Faculty of Pharmacy, Heliopolis University, Cairo 11785, Egypt; 5Department of Pharmaceutical Chemistry, Faculty of Pharmacy, Nahda University, Beni-Suef 62521, Egypt; mostafa.mansour@nub.edu.eg; 6Department of Medicinal Chemistry and Pharmacognosy, College of Pharmacy, Qassim University, Buraidah 51452, Saudi Arabia; s.almahmoud@qu.edu.sa; 7Department of Pharmaceutical Chemistry, College of Pharmacy, University of Hail, Hail 81442, Saudi Arabia; b.huwaimel@uoh.edu.sa

**Keywords:** phytochemical study, molecular docking, dynamic simulation, ADMET, CDK-2/6, topoisomerase-1, VEGFR-2

## Abstract

Genus *Tabebuia* is famous for its traditional uses and valuable phytoconstituents. Our previous investigation of *Tabebuia* species noted the promising anticancer activity of *T. guayacan* Hemsl. leaves extract, however, the mechanism underlying the observed anticancer activity is still unexplored. The current research was designed to explore the phytochemical content as well as to address the phytoconstituent(s) responsible for the recorded anticancer activity. Accordingly, sixteen compounds were isolated, and their structures were elucidated using different spectroscopic techniques. The drug-likeness of the isolated compounds, as well as their binding affinity with four anticancer drug target receptors: CDK-2/6, topoisomerase-1, and VEGFR-2, were evaluated. Additionally, the most promising compounds were in vitro evaluated for inhibitory activities against CDK-2/6 and VEGFR-2 enzymes using kinase assays method. Corosolic acid (**3**) and luteolin-7-*O*-β-glucoside (**16**) were the most active inhibitors against CDK-2 (−13.44 kcal/mol) and topoisomerase 1 (−13.83 kcal/mol), respectively. Meanwhile, quercetin 3-*O*-β-xyloside (**10**) scored the highest binding free energies against both CDK-6 (−16.23 kcal/mol) as well as against VEGFR-2 protein targets (−10.39 kcal/mol). Molecular dynamic simulation indicated that quercetin 3-*O*-β-xyloside (**10**) exhibited the least fluctuations and deviations from the starting binding pose with RMSD (2.6 Å). Interestingly, in vitro testing results confirmed the potent activity of **10** (IC_50_ = 0.154 µg/mL) compared to IC_50_ = 0.159 µg/mL of the reference drug ribociclib. These findings suggest the three noted compounds (**3**, **10,** and **16**) for further in vivo anticancer studies.

## 1. Introduction

According to the “International Agency for Research on Cancer“, cancer represents the leading cause of death and an important barrier for increasing life all over the world, with approximately 19.3 million new cases and 10.0 million deaths from cancer in 2020. Thus, development of new, safe, and more specific biological targets is critical, particularly for the most aggressive tumors [1]. Oncogenic proteins represent a promising strategy in the development of new anticancer agents. More than 40 kinase inhibitors have been approved by FDA for treating different types of cancers [2].

Cyclin-dependent kinase inhibitors (CDKIs) are paid much attention, due to their important role in cell division and differentiation. Among the CDK family, CDK-2 and CDK-6 play a vital role in cells progression from G1 to S cell cycle phases [3,4]. Overexpression of CDK-2 was reported in several solid tumors and is also associated with radiotherapy resistance [2]. CDK-6 overexpression was reported in different types of lymphoma and breast cancer [5]. Consequently, CDKIs play a significant role in cancer treatment, preventing or at least reducing therapeutic resistance mechanisms [6].

Vascular endothelial growth factor (VEGF) is another oncogenic protein that acts as a positive regulator for vascular endothelial cells. VEGFR-2 mediate the proliferation, differentiation, and microvascular permeability of endothelial cells [7]. VEGFR-2 is overexpressed in several malignancies; thus, blocking the VEGFR-2 pathway can change or destroy tumor vessels, and already many in vitro VEGFR-2 inhibitors have achieved clinical success in cancer treatment [8].

Furthermore, a significant increase of topoisomerase-1 was detected in surgical specimens of colon adenocarcinoma, non-Hodgkin’s lymphoma, leukemia, stomach, breast, lung, and malignant melanoma. Drug inhibitors caused paralyzing of topoisomerase-1 cleavage complexes, leading to DNA damage and cancer cell death [9].

Natural products are the most important safe source for discovering anticancer candidates. More than 3000 plants have been reported to have anticancer activity [10].

Genus *Tabebuia* has been used for a long time as a therapeutic alternative by rural populations, especially in Colombia, Brazil, and Latin American countries [11]. The FDA registered *T. impetigionosa* bark tea as a dietary supplement to alleviate conditions and symptoms associated with cancer [12]. In addition, *Tabebuia* species are widely used in the treatment of several ailments, such as syphilis, malaria, stomach disorders, inflammation, bacterial and fungal infections, poor memory, irritability, and depression [13]. Furthermore, β-lapachone, a naphthoquinone found in most *Tabebuia,* species, is now in the clinical trial and drug development phase as a plant-derived anticancer agent [14].

According to El-Hawary et al., *T. guaycane* Hemsl. is inadequate chemically and pharmacologically investigated. Previous phytochemical study of the plant bark led to the isolation of 6 naphthoquinone compounds (lapachol, α-lapachone, dehydro-α-lapachone, β-lapachone, Guayin, and Guayacanin) [15], while no previous reports were found discussing the phytochemical content of the leaves part. We previously investigated the cytotoxic potential of some *Tabebuia* species, and the results indicated cytotoxic activity of *T. guayacan* Hemsl. leaves extract against the two cancer cell lines HepG2 and Caco2 with IC_50_ 13.4 µg/mL and 12.2 µg/mL, respectively [16]. Accordingly, the current study was designed to investigate the phytochemical content of this species, followed by the evaluation of the cytotoxic potential of the isolated metabolites using molecular docking and dynamic simulation techniques, as well as in vitro testing of the enzyme inhibitory activity of the most promising compounds.

Molecular docking is a theoretical simulation that studies the interaction between molecules and predicts the orientation of the ligand while binding to a protein receptor using electrostatic interactions, and the sum of interactions is approximated by a docking score, which represents the potentiality of binding [17]. Dynamic simulations were performed to check the stability of the docked complexes taking into consideration the effect of the biological system [18].

In the current study, all isolated constituents were molecularly docked against the four oncogenes: CDK-2/6, topoisomerase-1, and VEGFR-2, using ribociclib as a positive control for CDK-2/6 targets [19], irinotecan, as a positive control for topoisomerase-1 [20], and sorafenib as a positive control for VEGFR-2 [21]. The chemical structures for the three positive controls are shown in Figure 1.

The pharmacokinetic properties have also been studied. To achieve depth analysis, we extended our study by performing the molecular dynamic simulation of 50 ns, and the most active constituents in the docking study were then analyzed using both the root mean square deviation (RMSD) and binding energy (Δ*G*). The most active constituents were also evaluated for their in vitro inhibitory activities against the studied oncogenes.

## 2. Results

### 2.1. Compounds Identification

The chromatographic investigation of various fractions (DCM, EtOAc, and *n*-butanol) from *T. guayacan* Hemsl. leaves extract yielded sixteen compounds **1**–**16** (Figure 2). The isolated compounds were identified using a variety of spectroscopic methods, including: UV, HRMS, ^1^H-NMR, and DEPT-Q NMR analyses as well as Co-TLC with authentic samples, and data comparison with the published literature. The isolated compounds were identified as; β-sitosterol (**1**) [22], ursolic acid (**2**) [23,24], corosolic acid (**3**) [25], 3-*O*-*trans-p*-coumaroylcrosolic acid (**4**) [23,26], 3,6,19 trihydroxy-ursolic acid (**5**) [27], β-sitosterol 3-*O*-β-glucoside (**6**) [28], quercetin (**7**) [29], luteolin (**8**) [30], quercetin-3-*O*-glucoside (**9**) [31], quercetin 3-*O*-*β*-xyloside (**10**) [32,33],4-hydroxybenzoic acid (**11**) [34], 4-methoxy benzoic acid (**12**) [34], 3,4-dihydroxybenzoic acid (**13**) [35], *p*-coumaric acid (**14**) [36], rutin (**15**) [37], and luteolin-7-*O*-β-glucoside (**16**) [38].

### 2.2. Docking Study

The docking study was carried out to identify the possible interactions between the ligand and the biological target. The positive control drug of each target was redocked into the active site. In order to allow the comparison of affinity between the tested compounds and the positive control drugs in a more precise and effective way, RMSD and the binding free energy for each complex were calculated. RMSD values from 0.3698–0.8515 Å (Table 1) indicated the high reliability of MOE dock software in reproducing the binding affinity for these inhibitors.

Each of the 16 isolated secondary metabolites was docked inside the active site of each of the four different targets. Their binding scores were recorded in order to find out the most appropriate candidates compared to the positive control ligands (ribociclib, irinotecan, and sorafenib) (Table 2). The results revealed that all metabolites showed binding affinity to the four targets binding pockets with energy scoring ranging from −5.69 to −13.44 kcal/mol with CDK-2 target, −5.58 to −16.23 kcal/mol with CDK-6 target, while the docking scores to topoisomerase-1 target were from −5.55 to −13.83 kcal/mol and to VEGFR-2 target from −5.13 to −10.39 kcal/mol.

The triterpenoidal compound **3** and the steroidal compound **1** have the best docking affinity against CDK-2 ligand with binding energies −13.44 and −13.27 kcal/mol, respectively. These values were considered to be in a good range compared to their positive control (ribociclib) −17.58 kcal/mol (Figure 3, Appendix A).

Interestingly, the flavonoid glycoside **10** demonstrated a binding energy of −16.23 kcal/mol, against the CDK-6 active site, which was considered significant compared to the positive control ribociclib (−15.08 kcal/mol) (Figure 4, Appendix A). Furthermore, the same compound had the best fitting energy to the VEGFR-2 binding site (−10.39 kcal/mol), compared to sorafenib (−12.58 kcal/mol) (Figure 5, Appendix A).

Concerning topoisomerase-1, the best ligand binding affinity was achieved by the flavonoid glycoside **16** with binding free energies (−13.83 kcal/mol) compared to the referenced drug irinotecan (−14.65 kcal/mol) (Figure 6, Appendix A).

### 2.3. Molecular Dynamic Simulation

As the protein is considered rigid during the semi-flexible docking calculations and in order to get a more realistic picture between the protein and the ligands, the docked complexes were simulated in a water box for about 50 ns. The time evolution of RMSD was determined to check the structural stability of the protein, and the ligands during the simulation. For all the complexes, RMSDs profiles were determined for the protein, relative to the X-ray structure of the targeted CDK-2/6, topoisomerase-1, and VEGFR-2 proteins (PDB: 1DI8, 1XO2, 1T8I and 2OH4, respectively), as well as for the most active compounds (**3**, **10,** and **16**) and their referenced drugs (ribociclib, irinotecan, and sorafenib).

Further computational validation was also achieved through the calculation of the binding free energies (ΔG). The RMSD graph of the top-scoring compound **3**, with CDK-2 protein target (PDB ID: 1DI8) showed binding stability and ΔG value comparable to that of the reference inhibitor ribociclib over 50 ns of MDS. Both compounds exhibited average RMSD from the initial docking pose of 4.4 and 3.5 Å, respectively, and ΔG values of −7.9 and −7.4 kcal/mol, respectively; however, compound **3** showed significant fluctuation at 46 ns (Figure 7A).

Both compound **10** and the reference inhibitor ribociclib showed comparable binding stability (average RMSD ~2.6 and 2.2 Å, respectively) with CDK-6 protein target (Figure 7B). Moreover, they achieved close ΔG values (−7.9 and −8.0 kcal/mol, respectively). Compound **16** and the reference inhibitor (irinotecan) achieved convergent average RMSDs over the course of MDS (~2.6 and 2.1 Å, respectively; Figure 7C) with topoisomerase 1 protein target. However, **16** was significantly fluctuating in comparison to irinotecan, and hence, this was reflected in their ΔG values that are equivalent to −6.9 and −8.6 kcal/mol, respectively. Finally, compound **10** and the reference inhibitor sorafenib were also convergent in their binding stability inside the active site of the VEGFR-2 protein target (RMSD~2.9 and 2.1 Å, respectively), and they achieved ΔG values of −8.1 and −8.7 kcal/mol, respectively (Figure 7D).

### 2.4. In Vitro Enzyme Inhibition

Based upon the previous findings, the most active docked cytotoxic compounds (**1, 3, 10**) were selected to evaluate their in vitro enzymatic inhibitory activities, in which **1** and **3** were evaluated against CDK-2 protein kinase and **10** was evaluated against both CDK-6 and VEGFR-2 enzymes using the kinase assays method. The results were reported as a 50% inhibition concentration value (IC_50_ µg/mL), as shown in Table 3.

### 2.5. ADMET Properties Evaluation

Pharmacokinetic evaluation is an important step in drug discovery to optimize acceptable properties and low toxicity. The ADMET parameters of the most potent constituents (**1, 3, 10,** and **16**), including absorption, distribution, metabolism, excretion, and toxicity, were evaluated, and are illustrated in Table 4. The results showed that all compounds have molecular weights less than 500 Da, indicating that they are easily eliminated. Also, they exhibited good water solubility and skin permeability. The intestinal absorption was in the good range where compound **3** scored the highest intestinal absorption (100%).

The volume of distribution (VDss) was also acceptable where compound **10** exhibited a very-high-volume distribution. Only **1** can cross the BBB, additionally, **1** and **3** have CNS permeability. Cytochrome P450 enzymes contain more than 50 enzymes, six of them metabolize 90% of drugs, with the two most important enzymes being CYP3A4 and CYP2D6 [39]. Compound **16** can inhibit the CYP1A2 enzyme and may play a role in drug–drug interactions, all compounds are non-CYP isoform inhibitors. None of the compounds are mutagenic, carcinogenic, hepatotoxic, skin irritant, or minnow toxic. However, three of them (**1**, **10,** and **16**) showed partial cardiac toxicity (hERG II inhibitor) as well as *T. Pyriformis* toxicity.

## 3. Discussion

For a long time, genus *Tabebuia* has been used in traditional medicine. According to the most recent review [15], several *Tabebuia* species have not yet been phytochemically and/or biologically discovered. The findings of our previous research, discussing the cytotoxic activity of some *Tabebuia* species [16] encouraged us to further investigate the phytochemical content of *T. guayacan* Hemsl. and acknowledge the most promising cytotoxic constituents using docking and dynamic simulation techniques. Additionally, the in silico results were augmented by the results of in vitro enzyme inhibition testing, which confirmed the observed results.

Sixteen compounds were isolated and identified as: two sterols (**1 and 6**), four triterpenes (**2**–**5**), two flavonoids aglycon (**7–8**), four flavonoids glycoside (**9, 10, 15,** and **16**) and four phenolic acids (**11**–**14**).

Molecular docking technology is an important step for scientific research, as its use provides a potent computational filter in order to reduce work and cost required for effective drug development, along with its ability to give a good interpretation for bioactive mechanisms [40]. The relative fitting affinities of the sixteen isolated compounds toward the four targets are represented in Figure 8.

Compound **10** achieved an interesting binding score energy (−16.23 kcal/mol) compared to the positive control (−15.08 kcal/mol) against CDK-6 protein target (Table 2 and Figure 9A).

CDK-6 protein target, as other kinases, consists of two domains, the smaller N-terminal domain (residues 1–100) and the larger C-terminal domain (residues 101–308) [41]. In general, compound **10** binds more extensively to the N-terminal domain than to the C-terminal domain. For detailed analyses (Appendix A, Figure 4), the hydrophobic core of the benzopyran ring made numerous Van der Waals interactions with CDK-6 residues (Ile19-Val 27- Ala 41 and Val 77) located in the N-terminal domain, similarly with (Phe 98 and Ala 102) from the hinge area connecting the two CDK-6 domains together, and also with (Ala 162 and Leu 152) residues from the C-terminal domain. According to Khuntawee et al., Van der Waals interactions represent an important factor in the binding efficiency of flavonoids against CDK-6 target [42]. In addition, **10** was able to exhibit two H-bond interactions with Asp 163 CDK-6 residue; one of them is a strong interaction (2.9 Å), as Anne Imberty et al. showed that the distances of hydrogen bonds between 2.5 and 3.1 Å are considered a strong interaction [43]. Furthermore, Asp 163 residue is essential for the catalytic activity of eukaryotic protein kinases and is considered a crucial active residue [41]. Moreover, the two interactions of the di-hydroxy phenyl group in the ATP-binding pocket with (Phe 98) residue, made the drug a good inhibitor because competition with the ATP-binding pocket makes the kinase activity of the CDK-6/cycle D complex stop [44,45]. Conclusively, compound **10** achieved a very good fitting inside the enzyme active site by 3 H-donor and 9 hydrophobic interactions with the reported active amino acids residues, noting this compound as a future hope for a new selective CDK-6 inhibitor.

Compound **10** also scored the best binding affinity among the tested isolated compounds against VEGFR-2 protein target (Figure 9B). The compound was able to form two interactions with the VEGFR-2 catalytic site (Appendix A, Figure 5), one of them was H-donor with (Asp 1044) residue and the other was hydrophobic with (Glu 883) residue and both residues were considered crucial residues for activity. The lack of interaction against the other active residue (Cys 917) plus the lack of pharmacophore functional group as in sorafenib (urea) plays a vital role in limiting the binding affinity [46].

Human DNA topoisomerase-1 is a multi-domain enzyme which contains two highly conserved globular domains (the core and the COOH-terminal domain) that are crucial for catalytic activity. Staker et al., reported that the glucose moiety seems to be important for the binding affinity where the removal of the glucose moiety from some drug inhibitors decrease topoisomerase-1 inhibition and DNA intercalation [46]. This nicely explained the characteristic binding affinity observed for **16** and **9** against topoisomerase-1 protein target (Figure 9C).

Moreover, **16** shared the positive control (irinotecan) the presence of three interactions with (Glu 356, Asp 533, and Lys 532) topoisomerase-1 residues (Appendix A, Figure 6), which represent essential residues for activity [47,48]. However, irinotecan has a higher activity due to the presence of another interaction with the active residue (Thr 718) in addition to numerous H-bond and Van der Waals interactions with DNA basis and other active enzyme residues.

The structure of CDK-2 consists of an amino-terminal lobe and a carboxy-terminal lobe. ATP binds in a deep cleft between the two lobes which contain catalytic residues presumed to be the site of protein substrate binding and catalysis [49]. Compound **3** scored the best binding affinity to CDK-2 target as it exhibited 2 H-donor interactions with both Asp 145 and Ph 80 that are located in the ATP- binding active site [45], making **3** more potent than **1,** which misses Ph 80 active residue.

The cyclin A-CDK-2 interface contains approximately 25 hydrophobic residues versus 17 intermolecular hydrogen bonds [49]. This could explain the activity of the positive control (ribociclib), as well as the triterpenes (**2**–**5**) and sterol compounds (**1,6**) that were recorded as the more active among the 16 docked compounds against CDK-2 protein target (Figure 9D).

Among the three most promising compounds (**3, 10,** and **16**) compound **10** showed the least fluctuations and deviations from its starting binding pose with RMSD value equal to 2.6 Å and Δ*G* value of −7.9 kcal/mol. It also showed slight fluctuation at 12 ns that became equilibrated around this point until the end of MDS with very low fluctuations. According to [50], RMSD is in an accepted range between 2.0 and 3.0 Å. Additionally, Allam et al. stated that the compounds that can achieve ΔG of −7 kcal/mol or lower, have a high potential to be active in vitro [51]. This statement was nicely confirmed by the current findings of the in vitro enzyme inhibition testing of **10**. Furthermore, the results of the in vitro oncogenic protein inhibition investigation, (Table 3), were in a good agreement with the docking study, where compound **3** showed significantly higher activity (IC_50_ = 0.113 µg/mL) than **1** (IC_50_ = 0.241 µg/mL). Moreover, compound **10** displayed a strong activity (IC_50_ = 0.154 µg/mL) that was non-significantly different with the reference drug (ribociclib, IC_50_ = 0.159 µg/mL). This was also showed in the docking simulation, where **10** displayed binding free energy (−16.23 kcal/mol) close to ribociclib (−15.08 kcal/mol). It is worth noting that, this is the first report for in vitro testing of the inhibitory activity of compounds: **1**, **3,** and **10** against CDK-2 and CDK-6.

Previous investigation of the inhibitory activity of **16** against topoisomerase-1 [52,53] confirmed our results that **16** is especially poisonous to topoisomerase-1 protein target, suggesting this as the mechanism of this compound as a cytotoxic agent.

The prediction of the pharmacokinetic properties is an important step for drug discovery. Unfortunately, only 50% of reported cytotoxic drugs showed acceptable ADME properties, so the need to improve the ADME behaviors is necessary to avoid side effects and clinical failures [45]. Herein, sixteen compounds were isolated, four of which, (**1, 3, 10,** and **16**), were considered promising. The rest of the isolated compounds, on the other hand, were good candidates for drug beings. The majority of them, (except **4, 6**, and **15**), have a low molecular weight (less than 500 Da) indicating good elimination. Noting that absorbance value less than 30% indicates poor absorbance, all compounds showed good absorbance in the human intestine except for compound **15** and, with the exception of **11,** all compounds showed good skin permeability, indicating that they are valid for topical treatment. Glycoprotein (Pgp) extracts the foreign substances from the cell, and cancer cells often overexpress P-glycoprotein [54]. The majority of the isolated compounds were Pgp substrates, while some were not a substrate, such as compounds **1**–**3**, and **11**–**14**. In terms of metabolism, the majority of the compounds appeared to be non-CYP450 inhibitors. However, compound **7** inhibited both CYP1A2 and CYP3A4 enzymes, compound **8** inhibited both CYP2C9 and CYP3A4 enzymes, and compound **16** only inhibited CYP1A2 enzyme. It is noteworthy that none of the compounds are mutagenic, carcinogenic, hepatotoxic, or skin irritant.

Compound **10** scored the highest binding affinity (−16.23 kcal/mol) as well as a non-significant enzyme inhibitory activity (IC_50_ = 0.154 µg/mL) when compared to the synthetic drug (ribociclib, −15.08 kcal/mol, IC_50_ = 0.159 µg/mL, respectively). Ribociclib showed some clinical complications, some of which may influence the quality of life, such as diarrhea, hand and foot skin interaction, and fatigue, but others are fatal, such as cardiovascular events, arterial thromboembolic events, and bleeding [55]. In this regard, searching for natural alternatives represents a promising approach to decreasing toxicity.

The evaluation of the ADMET profile of **10** versus ribociclib (Table 4) led to several main findings. Ribociclib is a non-P-glycoprotein I inhibitor, so ribociclib may be involved in increasing the bioavailability of other drugs. Compound **10** showed higher VDss than ribociclib, making it more distributed to the tissues than the plasma, and this advantage is important for solid cancer tissues treatment. According to the toxicity profile, ribociclib is hepatotoxic, and the low results of both maximum tolerated dose and minnow toxicity indicated high acute toxicity.

On the other side, ribociclib has higher absorption than **10**, but now the application of nanotechnology leads to an increase in the bioavailability and the bioactivity of phytomedicine by aiding the target delivery [56].

## 4. Materials and Methods

### 4.1. Collection of Plant Material

The leaves of *T. guayacan* (Seem.) Hemsl. were collected during September 2020 from Al-Zohriya garden, Zamalek, Cairo Governorate, Egypt. The plant was kindly identified by Prof. Dr. Abdel-Halim Mohammed (Professor of Agriculture, Flora Department, Agricultural Museum, Dokki, Giza, Egypt). Voucher specimen kept in the Botanical garden in the Agricultural Museum, Dokki, Giza, Egypt with number (Ta 1). Plant leaves were washed with fresh water and dried in the shade with occasional sun for several days. The dried leaves were ground into a coarse powder by a grinding machine and stored at room temperature for the extraction process.

### 4.2. Chemicals

*n-*hexane (60–80 °C), methylene chloride (DCM), ethyl acetate (EtOAc), *n-*butanol, and methanol were purchased from El-Nasr Company for Pharmaceuticals and Chemicals, Egypt, and were distilled before use. Deuterated solvents (Sigma-Aldrich, Germany), including methanol (CD_3_OD), chloroform (CDCL_3_), and dimethyl sulfoxide (DMSO-*d6*), were used for nuclear magnetic resonance (NMR) spectroscopic analyses.

### 4.3. Chromatographic Materials

Silica gel G 60 for column chromatography (70–230 mesh) (Sigma-Aldrich, Germany), Sephadex LH 20 (Pharmacia, Uppsala, Sweden), Polyamide powder S6 for column chromatography (Riedel–De Haen AG, Seezle–Hannover, Germany), aluminum sheet (20 × 20 cm) precoated with silica gel 60 F254, (Merck, Darmstadt, Germany), *p*-anisaldehyde/H_2_SO_4_ spray reagent, and aluminum chloride spray reagent.

### 4.4. General Experimental Procedures

^1^H-NMR (400 MHz), and DEPT-Q NMR (100 MHz) spectra were recorded on a Bruker 400TM ASCEND NMR Spectrometer (NMR laboratory, Microanalytical unit) faculty of pharmacy, Beni-Suef University. Chemical shifts are presented in δ (ppm) using tetramethylsilane (TMS) as an internal standard and coupling constants (*J*) are expressed in Hertz (Hz). Detection of spots was observed under long and short wavelength UV light (Fisher Scientific LCF-445) at 366 and 254 nm. Melting points were measured using an Electrothermal IA9100 melting point apparatus (Stone, Staffordshire, ST15 OSA, UK).

### 4.5. Extraction and Isolation Procedure

The dried powdered *T. guayacan* Hemsl. leaves (2.5 kg) were extracted with 70% ethanol by cold maceration to exhaustion, evaporated under reduced pressure by vacuum distillation at a temperature not exceeding 40 °C. The residue (240 g) was saved for successive liquid–liquid fractionation. The dried ethanolic residue was suspended in the least amount of distilled water (200 mL) and subjected to liquid–liquid fractionation with *n*-hexane, methylene chloride (DCM), ethyl acetate (EtOAc), and *n-*butanol. The solvent in each case was evaporated and the dried solvent-free successive fractions were weighted (40, 25, 20, 30 g respectively) and saved for further examination and isolation. The extracts were triturated with *n*-hexane to get the fat-free extract. The methylene chloride fraction (25 g) was chromatographed over a glass column chromatography packed with 400 g silica gel 60. Gradient elution was carried out with 100% *n*-hexane then with *n*-hexane containing 5% stepwise increments of DCM up to 100% DCM, then polarity increased with EtOAc up to 100% EtOAc to give 60 fractions each 100 mL; all fractions were collected and pooled for TLC analysis to afford 8 sub-fractions (M1-M8) using the solvent systems (*n*-hexane/EtOAc 9.5:0.5- *n*-hexane/EtOAc 9:1- *n*-hexane/EtOAc 8:2- DCM/methanol (MeOH) 9.5:0.5- DCM/MeOH 9:1 and DCM/MeOH 8.5:1.5). The second sub-fraction M2 (1.4 g) was fractionated with DCM: EtOAc (95% DCM to 100% EtOAc) and further purified with *n*-hexane: EtOAc with a gradual increase in polarity by 2% which resulted in the separation of compounds **1**. The third group M3 (3.0 g) was fractionated with DCM: EtOAc (5% increment) and purified with DCM: MeOH with a gradual increase in polarity by 2%, then the fraction was further purified using RP 18 column chromatography with 100% MeOH which resulted in the separation of compounds: **2** and **3**. M5 (2 g) was fractionated with DCM: EtOAc and further purified with DCM: MeOH to give compounds **4** and **5**. Finally, the last subfraction M8 (2.5 g) was fractionated with DCM: EtOAc (5% increment) and further purified with DCM: MeOH to give compound **6**. The ethyl acetate extract (20 g) was chromatographed over a glass column packed with 450 g silica gel 60. Gradient elution was carried out with 100% DCM then the polarity increased with EtOAc up to 100% EtOAc and, finally, 5% stepwise increments of MeOH to give 80 fractions each 100 mL. All fractions were collected and pooled for TLC analysis to afford 10 subfractions (E1-E10) using the solvent systems (DCM/MeOH 9.5:0.5- DCM/MeOH 9:1, DCM/MeOH 8:2 and EtOAc/acetic acid/formic acid/H_2_O 100:11:11:27). E3 (2.5 g) was chromatographed on a silica gel and eluted with DCM: EtOAc to be further purified by column chromatography with a mixture of DCM: MeOH (2% increment) to give compound **7**. E4 and E5 were chromatographed in the same manner, eluted with EtOAc: MeOH (5% increment), and further purified DCM: MeOH (2% increment) to give compounds **8** and **9** from E4, and **10** and **11** from E5. E8 was eluted with EtOAc: MeOH (up to 100% EtOAc) to obtain a sub-fraction which was further chromatographed by a mixture of DCM: MeOH (4:6) up to 100% MeOH and then purified using a Sephadex column MeOH-H_2_O (80:20) to give compounds **12** and **13**. E10 was eluted with EtOAc: MeOH (2:8) and then purified with a Sephadex column MeOH-H_2_O (80:20) to give compound **14**. Finally, the *n*-butanol fraction (30 g) was chromatographed over a column packed with 250 g polyamide. Gradient elution was carried out with 100% H_2_O then with water containing 5% stepwise increments of MeOH up to 100% MeOH, to give 7 subfractions (B1–B7) each 250mL, which were monitored by TLC on precoated silica gel plates using the solvent systems (DCM: MeOH 8:2, DCM: MeOH 7:3 and EtOAc—acetic acid– formic acid—H_2_O 100:11:11:27). B5 and B7 were further purified using a Sephadex column MeOH-H_2_O (80:20) to give compounds **15** and **16**, respectively.

### 4.6. Docking Study

To study the protein-ligand interactions, isolated compounds from the leaves of *T. guayacan* were sketched using Marvin Sketch powered by Chem-Axon and ChemBioDraw Ultra 14.0, and then applied to a Molecular Operating Environment (MOE) platform to undergo energy optimization for each compound using the MMFF94x force field (with the gradient set to root mean square (RMS) 0.1 kcal mol−1). All molecular modeling calculations and docking studies were carried out using ‘Molecular Operating Environment 2020.0101′ software (MOE) of Chemical Computing Group Inc., on a Core i7 2.2 GHz workstation) running on a Windows 10 PC [57]. Visualization and generation of the 3D figures were performed using PyMOL 2.4 software.

The X-ray crystallographic structures of the four targets (CDK-2 [PDB ID: 1DI8], CDK-6 [PDB ID: 1XO2], topoisomerase-1 [PDB ID: 1T8I], VEGFR-2 [PDB ID: 2OH4]) were obtained from the RSCB protein data bank (http://www.rcsb.org/, accessed on 12 November 2021). Each target was prepared by removal of waters, residue chains, and ligands that are not involved in the binding, except topoisomerase 1; we kept all chains as the active pocket involved more than one chain interaction, then prepared using the quick preparation protocol in MOE with default options [41,44,58,59].

#### 4.6.1. Ligand Preparation

The docked compounds (**1**–**16**) and the positive controls for each target were prepared for docking by applying the following steps: 2D structures of the docked ligands were built using Marvin Sketch and copied to MOE, then 3D protonation of the structure, selecting the least energetic conformer after running conformational analysis using systemic search [57].

#### 4.6.2. Docking Method Validation

To ensure that the ligand orientations and positions obtained from the docking studies were likely to represent valid and reasonable potential binding modes of the inhibitors, the docking methods and parameters used were validated by redocking of the native ligand of each protein. The re-docked native ligands showed root mean square deviation (RMSD) ranges from 0.3698 to 0.8515 between the docked poses and the co-crystallized ligands for the four targets. Docking protocols are summarized in Table 1.

### 4.7. Molecular Dynamic Simulation

Molecular dynamic simulation (MDS) for the generated ligand-enzyme complexes was performed using the Nanoscale Molecular Dynamics (NAMD) 2.6 software [60], applying the CHARMM27 force field [61]. Hydrogen atoms were added to the protein structures using the psfgen plugin included in the Visual Molecular Dynamic (VMD) 1.9 software [62]. The whole generated systems were then solvated using water molecules (TIP3P) and 0.15 M NaCl. At first, the total energy of the generated systems was minimized and gradually heated to reach 310 K and equilibrated for 200 s. Subsequently, the MDS was continued for 50 ns, and the trajectory was stored every 0.1 ns and further analyzed with the VMD 1.9 software. The MDS output was sampled every 0.1 ns to calculate the RMSD. The parameters were prepared using the online software VMD Force Field Toolkit (ffTK) [62]. Binding free energies (ΔG) were calculated using the free energy perturbation (FEP) method [63]. The web-based software Absolute Ligand Binder [63] was used to generate the input files for the NAMD software which performed the simulations required for ΔGs calculations.

### 4.8. In Vitro Enzyme Inhibition

#### 4.8.1. In Vitro CDK-2 and VEGFR-2 Inhibitory Activity

The Assay Kits are designed to measure CDK2/CyclinA2 and VEGFR-2 (KDR) activities for screening and profiling applications, using Kinase-Glo^®^ MAX as a detection reagent. The CDK-2 Assay Kit comes in a convenient 96-well format, with enough purified recombinant CDK2/CyclinA2 enzyme, CDK substrate peptide, ATP, and kinase assay buffer for 100 enzyme reactions [64], while the VEGFR-2 Assay Kit comes in a convenient 96-well format, with enough purified recombinant VEGFR-2 (KDR) enzyme, PTK substrate, ATP, and kinase assay buffer for 100 enzyme reactions [65]. The assays were performed according to the protocols supplied from the CDK2 Assay kit #79599 and the VEGFR-2 Assay kit #40325. The CDK2/CyclinA2 and VEGFR-2 (KDR) activities at a single dose concentration of 10µM were performed, where the Kinase-Glo MAX luminescence kinase assay kit (Promega#V6071) was used. The compounds were diluted in 10% DMSO and 5 µL of the dilution was added to a 50 µL reaction so that the final concentration of DMSO was 1% in all of the reactions. All of the enzymatic reactions were conducted at 30 °C for 45 min. The 50 µL reaction mixture contained 40 mM Tris, pH 7.4, 10 mM MgCl2, 0.1 mg/mL BSA, 1 mM DTT, 10 mM ATP, (kinase substrate and the enzyme CDK2/CyclinA2 for CDK-2 assay, or PTK substrate and the enzyme VEGFR-2 (KDR) for VEGFR assay). After the enzymatic reactions, 50 µL of Kinase-Glo^®^ MAX Luminescence kinase assay solution was added to each reaction and the plates were incubated for 15 min at room temperature. Luminescence signal was measured using a Bio Tek Synergy 2 microplate reader.

#### 4.8.2. In Vitro CDK-6 Inhibitory Activity

The CDK-6 inhibitory activity assay was performed using the ADP-Glo™ Kinase assay kit (Promega, Catalog: V4489) according to the manufacturer’s instructions [66]. CDK6/Cyclin D3 was incubated with the substrate, compounds, and ATP in a final buffer of 25 mM HEPES (pH 7.4), 10 mM MgCl2, 0.01% Triton X-100, 100 μ mL−1 BSA, 2.5 mM DTT in a 384-well plate with the total volume of 10 μL, respectively. ADP-Glo™ kinase assay was performed in two steps once the kinase reaction had finished, after which 5 μL ADP-Glo Reagent was added to stop the kinase reaction and deplete the unconsumed ATP. Only ADP and a very low background of ATP were left. The mixture was then incubated at room temperature for 40 min and 10 μL of Kinase detection reagent was added to transform ADP to ATP and introduced luciferase and luciferin to detect ATP. Finally, the mixture was incubated at room temperature for 30 min, before the luminescence was measured using a plate-reading luminometer. The signal was proportional to the amount of ATP present in the reaction and inversely proportional to the kinase activity.

#### 4.8.3. Statistical Analysis

The IC_50_ values of the most active compounds were expressed as mean ± SD (Table 3) and were calculated using GraphPad Prism 6 software (San Diego, CA, USA). Data were analyzed using one-way analysis of variance (ANOVA) for multiple comparison between all compounds followed by Tukey–Kramer post-ANOVA test. The differences were considered statistically significant at *p* < 0.05.

### 4.9. Prediction of the Pharmacokinetic Properties and Toxicological Properties Using ADMET

For the calculation of the pharmacokinetic properties for the most potent active constituents, online pkCSM pharmacokinetics prediction properties were used. The following properties: (i) absorption: water solubility, Caco2 permeability, intestinal absorption (human), skin permeability, and P-glycoprotein interactions; (ii) distribution: VDss, Fu, Log BB, and CNS permeability; (iii) metabolism; and (iv) excretion were selected. Additionally, online pkCSM pharmacokinetics were used to predict the toxicity of the molecules, including skin sensitization, hepatotoxicity, mutagenicity, and others. The results obtained were analyzed and compared with the reference values of pkCSM pharmacokinetics prediction properties [67].

## 5. Conclusions

The phytochemical study of the total ethanolic extract of *T. guayacan* Hemsl. leaves led to the isolation of sixteen compounds belonging to different classes of secondary metabolites. It is noteworthy that this is the first report for isolation of these compounds from *T. guayacan* Hemsl. Molecular docking investigations of the isolated phytochemicals were robustly executed with different oncogenes that have been reported to be actively involved in various forms of carcinoma. MDS and ADMET profiling were also performed. Compounds **3**, **10**, and **16** were the most active inhibitors, of which, compound **10** scored the best binding fitting energies against two protein targets (CDK-6 and VEGFR-2) and the least fluctuation and deviation in MDS technique. In vitro enzyme inhibition studies confirmed the previous results, noting compound **10** as a promising candidate. These findings provided valuable information about the potential anticancer activities of the phytoconstituents extracted from *T. guayacan* Hemsil. leaves.

## Figures and Tables

**Figure 1 plants-11-00888-f001:**
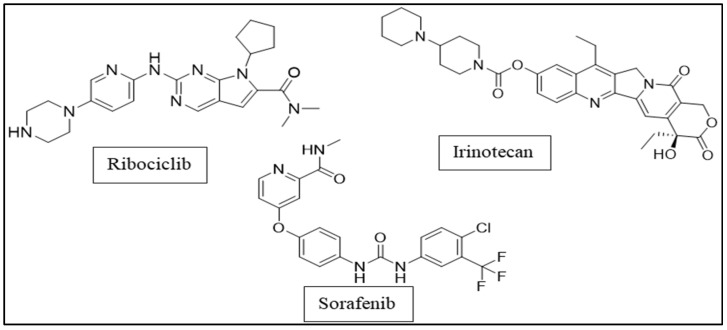
The chemical structures of the positive controls.

**Figure 2 plants-11-00888-f002:**
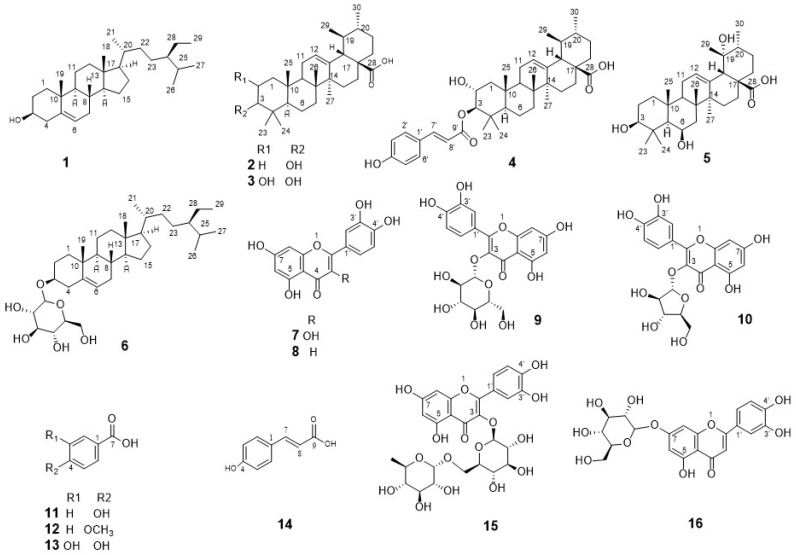
The chemical structures of the isolated compounds from *T. guayacan* Hemsl.

**Figure 3 plants-11-00888-f003:**
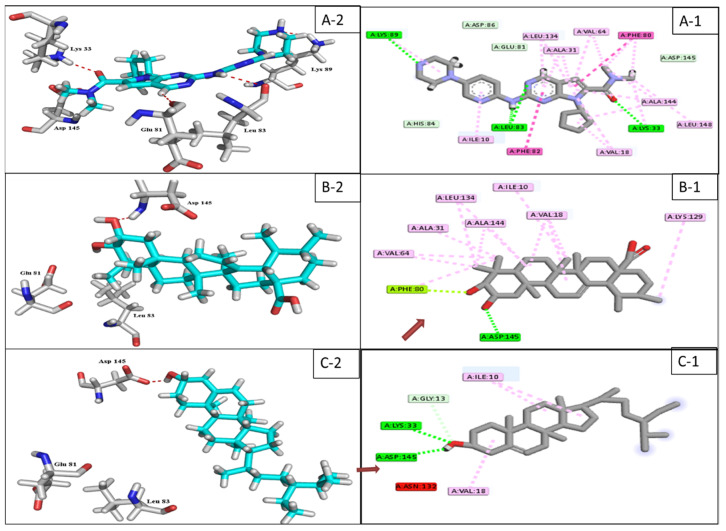
2D & 3D interaction of the top docking pose of ribociclib (**A-1**,**A-2**), compound **3** (**B-1**,**B-2**) and compound **1** (**C-1**,**C-2**) at the catalytic domain of CDK-2 enzyme.

**Figure 4 plants-11-00888-f004:**
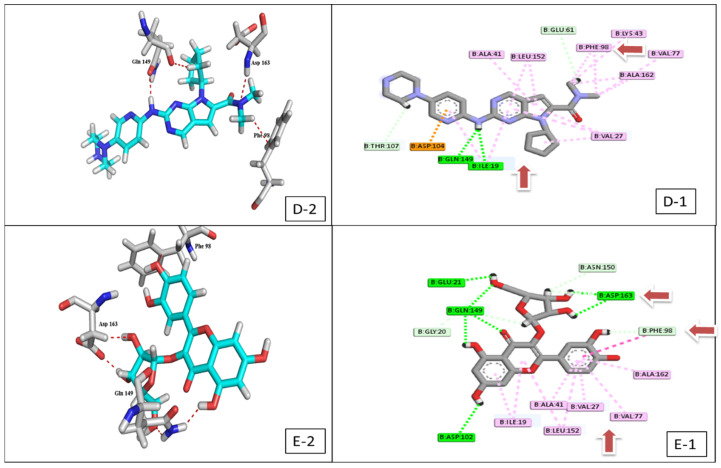
2D & 3D interaction of the top docking pose of ribociclib (**D-1**,**D-2**) and compound **10** (**E-1**,**E-2**) at the active domain of CDK-6 enzyme.

**Figure 5 plants-11-00888-f005:**
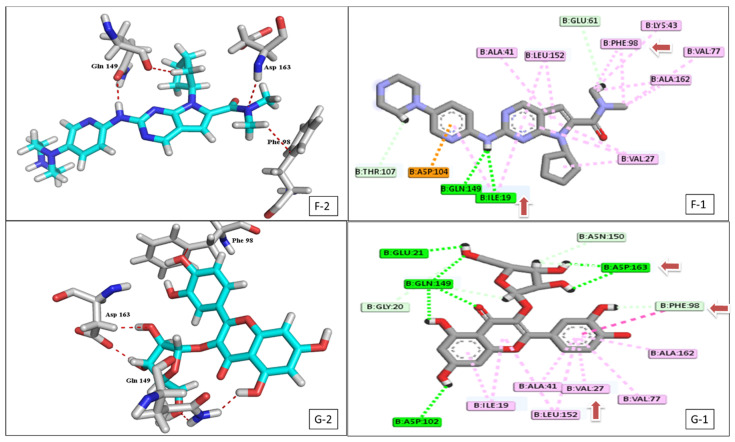
2D & 3D interaction of the top docking pose of sorafenib (**F-1**,**F-2**) and compound **10** (**G-1**,**G-2**) at the active site of VEGFR-2.

**Figure 6 plants-11-00888-f006:**
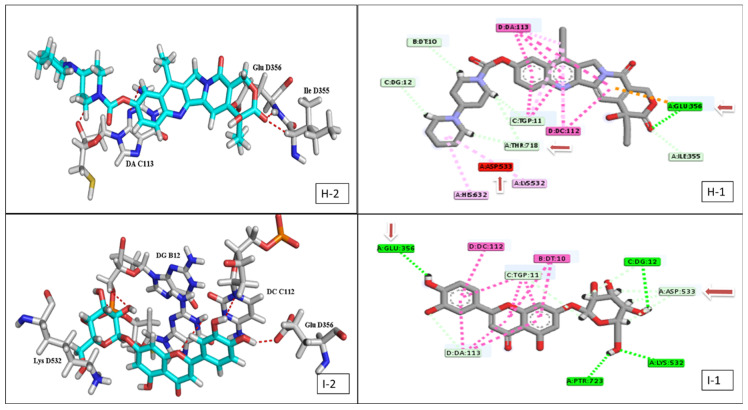
2D & 3D interaction of the top docking pose of irinotecan (**H-1**,**H-2**) and compound **16** (**I-1**,**I-2**) at the active site of topoisomerase-1 enzyme.

**Figure 7 plants-11-00888-f007:**
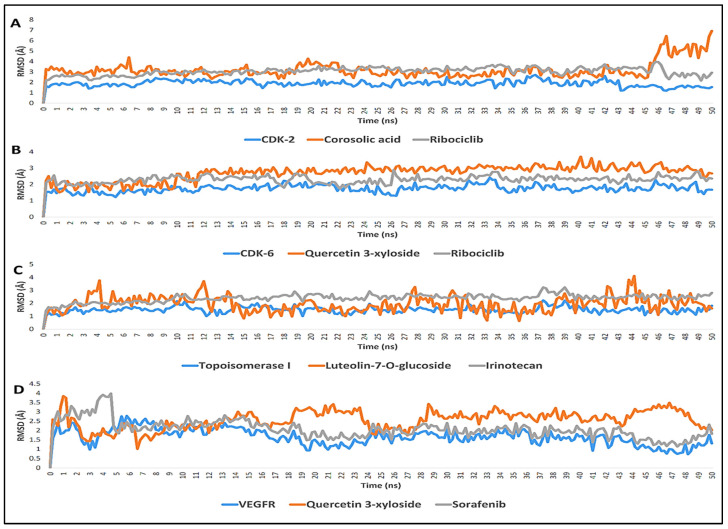
(**A**) The RMSDs of compound **3** inside the binding sites of CDK-2 target over 50 ns of MDS, (**B**) The RMSDs of compound **10** inside the binding sites of CDK-6 target over 50 ns of MDS, (**C**) The RMSDs of compound **16** inside the binding sites of topoisomerase-1 target over 50 ns of MDS, and (**D**) The RMSDs of compound **10** inside the binding sites of VEGFR-2 target over 50 ns of MDS.

**Figure 8 plants-11-00888-f008:**
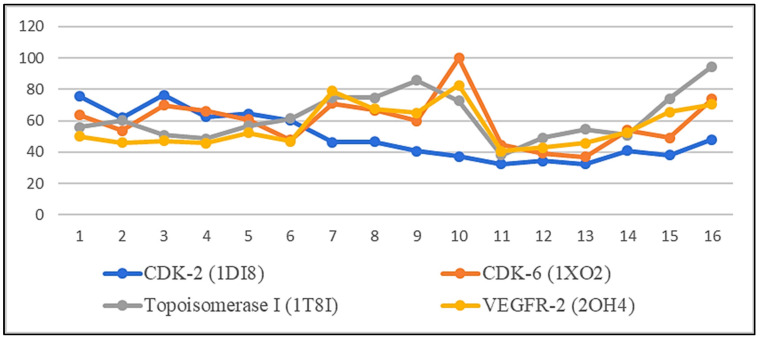
The relative percentage of the binding affinity of each isolated compound toward the four oncogenic targets.

**Figure 9 plants-11-00888-f009:**
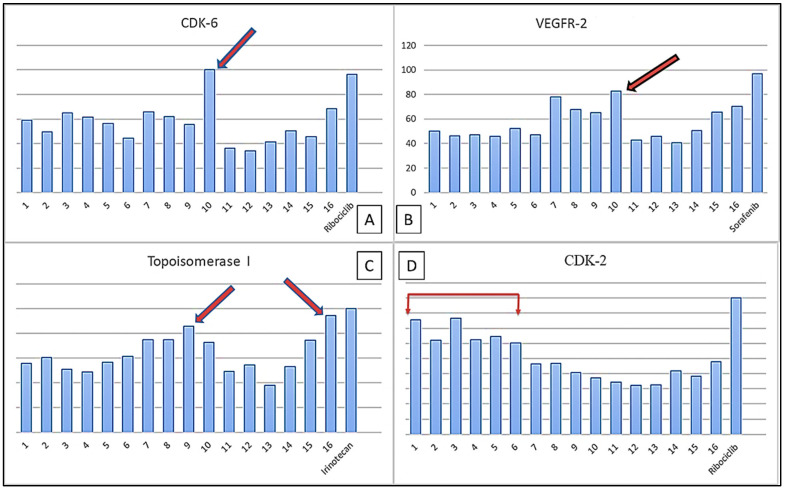
(**A**) The relative percentage of the binding affinity of quercetin-3-*O*-xyloside compared to other isolated constituents against CDK-6 target; (**B**) The relative percentage of the binding affinity of quercetin-3-*O*-xyloside and other isolated constituents against VEGFR-2 target; (**C**) The relative percentage of the binding affinity of both luteolin-7-*O*-glucoside and quercetin-3-*O*-glucoside compared to other isolated constituents against topoisomerase 1 target; (**D**) The relative percentage of the binding affinity of the triterpenoidal and steroidal compounds and other isolated constituents against CDK-2 target.

**Table 1 plants-11-00888-t001:** Docking protocols for each macromolecule target.

Macromolecule Target	PDB ID	RMSD of Validation	Initial Scoring Method	Final Scoring Method	Ligand Placement Method	Docking Protocol	Positive Control
CDK-2	1DI8	0.6897	London dG	GBVI/WSA dG	Triangle Matcher	Rigid receptor	Ribociclib
CDK-6	1XO2	0.3698	London dG	GBVI/WSA dG	Triangle Matcher	Rigid receptor	Ribociclib
Topoisomerase-1	1T8I	0.8515	London dG	London dG	Template Plugin Feature	Rigid receptor	Irinotecan
VEGFR-2	2OH4	0.4368	London dG	GBVI/WSA dG	Triangle Matcher	Rigid receptor	Sorafenib

**Table 2 plants-11-00888-t002:** The binding energies of the secondary metabolites docked with the four target macromolecules.

No.	Compound Name	CDK-2 (1DI8)	CDK-6 (1XO2)	Topoisomerase-1 (1T8I)	VEGFR-2 (2OH4)
1	*β*-Sitosterol	**−13.27**	−9.61	−8.17	−6.30
2	Ursolic acid	−10.88	−8.08	−8.84	−5.80
3	Corosolic acid	**−13.44**	−10.54	−7.44	−5.92
4	3-*O*-*p*-coumaroyl corosolic acid	−10.99	−9.98	−7.10	−5.75
5	3β-6β-19α-trihydroxy-urs-12-en-28-oic acid	−11.35	−9.18	−8.31	−6.57
6	*β*-Sitosterol-3-*O*-D-glucopyranoside	−10.61	−7.20	−8.99	−5.90
7	Quercetin	−8.13	−10.70	−10.96	−9.94
8	Luteolin	−8.22	−10.05	−10.93	−8.51
9	Quercetin -3-*O*-glucoside	−7.14	−9.03	−12.56	−8.19
10	Quercetin 3-*O*-xyloside	−6.54	**−16.23**	−10.66	**−10.39**
11	4-Hydroxybenzoic acid	−6.05	−5.90	−7.18	−5.39
12	4-Methoxybenzoic acid	−5.69	−5.58	−7.98	−5.76
13	3,4-Dihydroxybenzoic acid	−5.70	−6.73	−5.55	−5.13
14	*p*-coumaric acid	−7.35	−8.22	−7.80	−6.77
15	Rutin	−6.71	−7.42	−10.87	−8.25
16	Luteolin-7-*O*-glucoside	−8.43	−11.14	**−13.83**	−8.85
**Reference Ligands**		**(−17.58)** **Ribociclib**	**(−15.08)** **Ribociclib**	**(−14.65)** **Irinotecan**	**(−12.58)** **Sorafenib**

All values of binding energy are in kcal/mol, Bold values represent the most active constituents and their positive controls.

**Table 3 plants-11-00888-t003:** The IC_50_ of the most active cytotoxic compounds in µg/mL.

	CDK-2 IC_50_ (µg/mL)	CDK-6 IC_50_ (µg/mL)	VEGFR-2 IC_50_ (µg/mL)
Ribociclib	0.039 ± 0.002		
Compound **1**	0.241 ± 0.015 ^a^		
Compound **3**	0.113 ± 0.007 ^a,b^		
Ribociclib		0.159 ± 0.008	
Compound **10**		0.154 ± 0.007	
Sorafenib			0.039 ± 0.002
Compound **10**			0.084 ± 0.003 ^c^

Data in the table represent mean ± standard deviation (SD) where; ^a^ significantly different from positive control (Ribociclib), ^b^ significantly different from compound **1**, ^c^ significantly different from positive control (Sorafenib) at *p* < 0.05.

**Table 4 plants-11-00888-t004:** The ADMET analysis of the most active isolated compounds.

Property	Model Name	Predicted Value					Unit
		Compound 1	Compound 3	Compound 10	Compound 16	Sorafenib	
**Absorption**	Water solubility	−6.773	−3.04	−2.903	−3.325	**−4.255**	Numeric (log mol/L)
	Caco2 permeability	1.201	0.641	0.052	0.432	**0.762**	Numeric (log Papp in 10^−6^ cm/s)
	Intestinal absorption (human)	94.464	100	51.884	46.308	**85.494**	Numeric (% Absorbed)
	Skin Permeability	−2.783	−2.735	−2.735	−2.735	**−2.74**	Numeric (log Kp)
	P-glycoprotein substrate	No	No	Yes	Yes	**Yes**	Categorical (Yes/No)
	P-glycoprotein I inhibitor	Yes	No	No	No	**Yes**	Categorical (Yes/No)
	P-glycoprotein II inhibitor	Yes	No	No	No	**Yes**	Categorical (Yes/No)
**Distribution**	VDss (human)	0.193	−1.282	1.508	−0.106	**−0.009**	Numeric (log L/kg)
	Fraction unbound (human)	0	0.037	0.134	0.064	**0**	Numeric (Fu)
	BBB permeability	0.781	−0.473	−1.473	−1.61	**−1.473**	Numeric (log BB)
	CNS permeability	−1.705	−1.507	−4.215	−4.67	**−2.025**	Numeric (log PS)
**Metabolism**	CYP2D6 substrate	No	No	No	No	**No**	Categorical (Yes/No)
	CYP1A2 inhibitor	No	No	No	Yes	**No**	Categorical (Yes/No)
	CYP2C19 inhibitor	No	No	No	No	**Yes**	Categorical (Yes/No)
	CYP2C9 inhibitor	No	No	No	No	**Yes**	Categorical (Yes/No)
	CYP2D6 inhibitor	No	No	No	No	**No**	Categorical (Yes/No)
	CYP3A4 inhibitor	No	No	No	No	**Yes**	Categorical (Yes/No)
**Excretion**	Total Clearance	0.628	0.093	0.364	0.687	**−0.213**	Numeric (log mL/min/kg)
	Renal OCT2 substrate	No	No	No	No	**No**	Categorical (Yes/No)
**Toxicity**	AMES toxicity	No	No	No	No	**No**	Categorical (Yes/No)
	Max. tolerated dose (human)	−0.621	0.124	0.494	0.765	**0.253**	Numeric (log mg/kg/day)
	hERG I inhibitor	No	No	No	No	**No**	Categorical (Yes/No)
	hERG II inhibitor	Yes	No	Yes	Yes	**Yes**	Categorical (Yes/No)
	Oral Rat Acute Toxicity (LD50)	2.552	2.513	2.585	2.54	**2.14**	Numeric (mol/kg)
	Hepatotoxicity	No	No	No	No	**Yes**	Categorical (Yes/No)
	Skin Sensitization	No	No	No	No	**No**	Categorical (Yes/No)
	T.Pyriformis toxicity	0.43	0.285	0.285	0.285	**0.307**	Numeric (log ug/L)
	Minnow toxicity	−1.802	0.276	5.071	1.266	**−0.515**	Numeric (log mM)

## Data Availability

All data generated or analyzed during this study are included in this published article.

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
