# Peer review of "Characterization of Promising Cytotoxic Metabolites from Tabebuia guayacan Hemsl.: Computational Prediction and In Vitro Testing"

_plants, 2022, doi:10.3390/plants11070888_

Round 1

Reviewer 1 Report

In the paper submitted to me for review, the isolation of as many as 16 compounds from dried Tabebuia guayacan leaves was described for the first time. All these compounds were identified. Neither of them turned out to be a new structure. However, the described method of isolation using various and complex chromatographic methods is highly appreciated. The reviewed work is interesting and generally well prepared. All chemical, biological and computational experiments are supported by appropriate experimental material. And despite the very good overall impression after reading the whole text carefully, the first impression after reviewing it briefly raised some of my reservations.

First, is an exceptionally large number of references, rather typical for review papers. I believe that many of them could be abandoned without significantly affecting on the overall level of assessed work. (In line 82 it is even stated that there is very little literature on this matter.)

Second, the summary seems a bit too extensive and detailed.

Third, and perhaps most importantly, all compounds isolated are well-known substances, as can be seen in reference 37-75, quoted in lines 416-422. Therefore, I believe that it is unnecessary to provide full identification data (including spectral data). As a consequence, chapter 2.1. should be shortened to 1/2 page, where it would only be stated that the substances extracted from the plant were identified as "xxx" with data consistent with the reference "XX".

This is a place for another comment appears, of a more editorial nature. We usually write the formula for methylene chloride as CH2Cl2 and not "CH2CL2". This error occurs repeatedly in many workplaces, and the full name "Methylene chloride" is sometimes used to describe the chromatographic phases. I propose to insert the commonly used abbreviation "DCM" in all places, which will correspond well with the MeOH and EtOAc used.

And the next thing is Fig. 3-6. 2D models are very similar to 3D models. Wouldn't it be better to use simple, regular line structures for 2D models? (But it is not necessary.)

In the title of Tab 3, I propose to remove the word "Showed"

In Figure 8, on the vertical axis, there is no description and dimensioning of the scale and there is lack a reference substance in this plot.

And finally, the summarizing conclusion from the work indicating that among the isolated substances, compound 10 is the best, but can it be somehow related to its structure? Especially that among these substances there are others, with a very similar structure and significantly lower activity.

And there are all my comments. After correcting what is necessary and possible correction of other defects, I recommend that the above work can be accepted for publication in the "Plants" journal.

Author Response

Comments and Suggestions for Authors

In the paper submitted to me for review, the isolation of as many as 16 compounds from dried Tabebuia guayacan leaves was described for the first time. All these compounds were identified. Neither of them turned out to be a new structure. However, the described method of isolation using various and complex chromatographic methods is highly appreciated. The reviewed work is interesting and generally well prepared. All chemical, biological and computational experiments are supported by appropriate experimental material. And despite the very good overall impression after reading the whole text carefully, the first impression after reviewing it briefly raised some of my reservations.

  • First, is an exceptionally large number of references, rather typical for review papers. I believe that many of them could be abandoned without significantly affecting on the overall level of assessed work. (In line 82 it is even stated that there is very little literature on this matter.)

- Yes, there are a large number of references, but the majority of them are in the introduction and identification of compounds as there are identified compounds, but line 82 states that there are no previously reported phytochemical and biological studies concerning the leaves part of T. guayacane and our study is the first study to cover this point.

- We reduced the number of references from 102 to 67 without having a significant impact on the    overall level of assessed work.

2- Second, the summary seems a bit too extensive and detailed.

- The abstract is summarized.

3- Third, and perhaps most importantly, all compounds isolated are well-known substances, as can be seen in reference 37-75, quoted in lines 416-422. Therefore, I believe that it is unnecessary to provide full identification data (including spectral data). As a consequence, chapter 2.1. should be shortened to 1/2 page, where it would only be stated that the substances extracted from the plant were identified as "xxx" with data consistent with the reference "XX".

- This section is already present in the discussion part, and now is moved to the result part and all spectral analysis and NMR data are moved to the supplementary data.

4- This is a place for another comment appears, of a more editorial nature. We usually write the formula for methylene chloride as CH2Cl2 and not "CH2CL2". This error occurs repeatedly in many workplaces, and the full name "Methylene chloride" is sometimes used to describe the chromatographic phases. I propose to insert the commonly used abbreviation "DCM" in all places, which will correspond well with the MeOH and EtOAc used.

- Thank you for your suggestion, all "CH2CL2" is corrected to “DCM”.

5- And the next thing is Fig. 3-6. 2D models are very similar to 3D models. Wouldn't it be better to use simple, regular line structures for 2D models? (But it is not necessary.)

We provide the interactions represented in:

  1. i) 2D figures where are ligands appear in the stick model, not a 3D caption.
  2. ii) 3D figures to reveal the three-dimensional interaction between ligand and residues.

6- In the title of Tab 3, I propose to remove the word "Showed"

- The word is removed.

7- In Figure 8, on the vertical axis, there is no description and dimensioning of the scale and there is lack a reference substance in this plot.

- Figure (8) represent the relative percentage of activity of all identified compounds towards the four oncogenic targets in the docking study and the details of this figure are represented in figure (9) and detailed discussed from line 261 to line 317. The vertical axis is added and the caption of figure (8) is improved.

8- And finally, the summarizing conclusion from the work indicating that among the isolated substances, compound 10 is the best, but can it be somehow related to its structure? Especially that among these substances there are others, with a very similar structure and significantly lower activity.

- According to the binding energy score, compound 10 bearing xyloside moiety scored the best energy values as CDK-6 and VEGFR-2 inhibitor among all compounds, although other similar structures are not significant active (for example: compound 9) bearing glucoside moiety. Probably due to the size of the sugar ring, which contributed either to strong interactions at the binding site or the orientation of the ligand towards better binging at the active site, as SAR activity are detailed discussed in the discussion part.

- Take in consideration another reasons related to xylose or glucose unit its self. The cancer cells do not behave as normal cells in terms of glucose utilization. When the oxygen levels are high, normal cells direct the glucose to the mitochondria for oxidative phosphorylation, whereas cancer cells show increased uptake of glucose regardless of oxygen levels (Çoban et al. 2019)*. The effect of flavonoids as antioxidant and cytotoxic activity is related to that flavonoids can block the glucose uptake in myelocytic U937 cells. In addition, glucose analogue that is trapped inside the cells in the form of 2-deoxyglucose-6-phosphate, which cannot be metabolized further, resulting in lethality induced by energy crisis, is a strategy for cancer treatment (Aft et al. 2002)**, may explain the higher activity of flavonoids connected to sugar moiety (xylose) rather than glucose.

*Çoban, E. A., Tecimel, D., Åžahin, F., & Deniz, A. A. H. (2019). Targeting cancer metabolism and cell cycle by plant-derived compounds. Cell Biology and Translational Medicine, Volume 8, 125-134.‏

**Aft RL, Zhang FW, Gius D (2002) Evaluation of 2-deoxy-D-glucose as a chemotherapeutic agent: mechanism of cell death. Brit J Cancer 87(7):805–812

And there are all my comments. After correcting what is necessary and possible correction of other defects, I recommend that the above work can be accepted for publication in the "Plants" journal.

Reviewer 2 Report

The manuscript entitled "Characterization of promising cytotoxic metabolites from Tabebuia guayacan: computational prediction and in vitro testing" is well-written and -designed. Although the phytochemicals isolated and identified all are known, but described for the first time in this plant. The results are discussed well. I have some comments in order to improve the quality of manuscript:

Abstract

L25,26: the chromatographic techniques are not using to elucidate the chemical structures, please revise 

Introduction

L99,100: please revise, seems to be incomplete sentence

Figure 1: the structures must be drawn via ChemDraw or similar software, the current quality of structures is not acceptable

Results

L115: please revise "CH2CL2" to "CH2Cl2" throughout the text, check please the similar formula e.g. "CDCL3"

section 2.1: since all the compounds isolated are well-known and previously described, in my opinion the NMR spectra characteristics are not needed, I would recommend citing the proper references reporting those data for each comp. in the text, besides bringing the spectra as supplementary, as mentioned in the discussion section (it can be moved up to this section)

Figure 2, the quality should be improved (via ChemDraw, etc.)

Materials and methods

section 4.1. please add the voucher specimen code, etc.

please describe by which reason the ethanolic extract was utilized for the isolation procedure, why not the methanolic?

References

please check again the references and revise all according to the template

Author Response

Comments and Suggestions for Authors

The manuscript entitled "Characterization of promising cytotoxic metabolites from Tabebuia guayacan: computational prediction and in vitro testing" is well-written and -designed. Although the phytochemicals isolated and identified all are known, but described for the first time in this plant. The results are discussed well. I have some comments in order to improve the quality of manuscript:

Abstract

L25,26: the chromatographic techniques are not using to elucidate the chemical structures, please revise 

-The sentence modified.

Introduction

L99,100: please revise, seems to be incomplete sentence

-The sentence is completed.

Figure 1: the structures must be drawn via ChemDraw or similar software, the current quality of structures is not acceptable

-The structures in “figure 1” are drawn via the ChemDraw software.

Results

L115: please revise "CH2CL2" to "CH2Cl2" throughout the text, check please the similar formula e.g. "CDCL3"

-All "CH2CL2" is corrected to “DCM” which will correspond well with the MeOH and EtOAc used, as reviewer one recommended. "CDCL3" is checked.

section 2.1: since all the compounds isolated are well-known and previously described, in my opinion the NMR spectra characteristics are not needed, I would recommend citing the proper references reporting those data for each comp. in the text, besides bringing the spectra as supplementary, as mentioned in the discussion section (it can be moved up to this section)

-The discussion section that interpretate the compound’s identification is moved to the results part and all spectral analysis and NMR data are moved to the supplementary data.

Figure 2, the quality should be improved (via ChemDraw, etc.)

-The quality of figure 2 is improved.

Materials and methods

section 4.1. please add the voucher specimen code, etc.

-The code of voucher specimen is added.

please describe by which reason the ethanolic extract was utilized for the isolation procedure, why not the methanolic?

- Ethanol is favorable as it is safe for human consumption, a good solvent for extraction as methanol, and an ideal solvent for phenolic and polyphenolic extraction. Methanol is known poisonous and evaporates some very minute residue that may remain in the extract.

-As increasing the water concentration in the solvent enhances extraction yield, we used 70% ethanol in water for extraction.

Do, Q. D., Angkawijaya, A. E., Tran-Nguyen, P. L., Huynh, L. H., Soetaredjo, F. E., Ismadji, S., & Ju, Y. H. (2014). Effect of extraction solvent on total phenol content, total flavonoid content, and antioxidant activity of Limnophila aromatica. Journal of food and drug analysis22(3), 296-302.‏

References

please check again the references and revise all according to the template.

-All references are revised.

Reviewer 3 Report

Dear authors

Please fix these comments

1- Please include in the introduction parts some data related to the phytochemical composition of the tested plant species: Tabebuia guayacan

2- In results section, please improve the quality of all figures

3- Some figures (2D and 3D structures) are not complete: please try to fix this issue (especially figure 3)

4- Table 3: add statistical analysis between the mean values obtained

5- Please discuss the pharmacokinetic, druglikeness, medicinal properties and toxicity profiles of the identified compounds.

6- Please add (Hemsl.) in the tilte and in the main text.

Good Luck

Author Response

Please fix these comments

1- Please include in the introduction parts some data related to the phytochemical composition of the tested plant species: Tabebuia guayacan

The required data is included

2- In results section, please improve the quality of all figures

Figures quality is improved.

3- Some figures (2D and 3D structures) are not complete: please try to fix this issue (especially figure 3)

Figures were generated and captured by PyMOL 2.4 software. We focused on strongly interacted residues only and deleted the non-interacted residues to better explain the interactions with no confusion.  Figure 3 is improved.

4-Table 3: add statistical analysis between the mean values obtained

The statistical analysis is added to table 3 and clarified in the material and method part.

5-Please discuss the pharmacokinetics, drug-likeness, medicinal properties and toxicity profiles of the identified compounds.

This section is added to the discussion part and the ADMET analysis of identified compounds are added to supplementary data at Tables (S5 and S6)

6- Please add (Hemsl.) in the title and in the main text.

The word is added in the title and all main text.

Good Luck

Reviewer 4 Report

This study is the result of studying the Characterization of Promising Cytotoxic Metabolites of Tabebuia guayacan by Computational Prediction and In Vitro Testing. In particular, it was conducted in vitro studies fo CDK-2, CDK-6 and VEGFR-2 Inhibitory Activity. However, the manuscript is confusing and ideas are unorganized. Authors should consider correct it.
The title is too broad to be suitable for in vitro level studies.
Even in the text, animals or cells were not tested and words were selected from too broad a range. An example is "Pharmacokinetic Properties and Toxicological Properties". It tends to be too exaggerated to say that an ethanolic extract of T. guayacan was studied with the root mean square deviation (RMSD) and binding energy (ΔG). 
There are many problems associated with the new and lack of methodological details, raising concerns about the rigor of the project's experimental execution.

Author Response

Comments and Suggestions for Authors

This study is the result of studying the Characterization of Promising Cytotoxic Metabolites of Tabebuia guayacan by Computational Prediction and In Vitro Testing. In particular, it was conducted in vitro studies fo CDK-2, CDK-6 and VEGFR-2 Inhibitory Activity.

  • However, the manuscript is confusing and ideas are unorganized. Authors should consider correct it.

Thank you for your time and effort, and we would like to express the rational of our research as follow:

- First, the study is a continuation for our previous published study (El-Hawary et al., 2021) that explored the cytotoxic activity of T. guayacene against three cancer cell lines. The present study attempt to address the active constituent(s) that might be related to the recorded activity and by which oncogenic target and these ideas are clearly mentioned in the abstract and all manuscript in organized points.

- Second, Molecular docking technique is an important step for scientific researches, as their use provides a potent computational filter in order to reduce work and cost required for effective drug development, besides, their abilities to give a good interpretation for bioactive mechanisms [76]. This is what exactly happen in our research, the main findings resulted from both the docking and molecular dynamics studies were further confirmed by in vitro study and the manuscript recommended future in vivo studied for the most active constituents.

  • The title is too broad to be suitable for in vitro level studies.

The title is representing the size of work that exactly done in the manuscript.

  • Even in the text, animals or cells were not tested and words were selected from too broad a range. An example is "Pharmacokinetic Properties and Toxicological Properties". It tends to be too exaggerated to say that an ethanolic extract of T. guayacan was studied with the root mean square deviation (RMSD) and binding energy (ΔG). 

- Animals are not tested because it’s in vitro study, the cells tested before in our previously published work as I mentioned this study is a continuation study.

- “Pharmacokinetic Properties and Toxicological Properties” this sentence is used in the manuscript to describe the ADMET software study as I mentioned in the methods section, not describe the in vitro study.

  • There are many problems associated with the new and lack of methodological details, raising concerns about the rigor of the project's experimental execution.

- All methods, soft wares and techniques are mentioned in methods section.

Round 2

Reviewer 3 Report

Dear Authors

Please check again Figure 3.

Best regards

Reviewer 4 Report

The author tried to address the issues raised by this reviewer. As a result, the manuscript was improved.